# An integrative cross-omics analysis of DNA methylation sites of glucose and insulin homeostasis

Jun Liu ⓘ et al.[#]

Despite existing reports on differential DNA methylation in type 2 diabetes (T2D) and obesity, our understanding of its functional relevance remains limited. Here we show the effect of differential methylation in the early phases of T2D pathology by a blood-based epigenome-wide association study of 4808 non-diabetic Europeans in the discovery phase and 11,750 individuals in the replication. We identify CpGs in *LETM1*, *RBM20*, *IRS2*, *MAN2A2* and the 1q25.3 region associated with fasting insulin, and in *FCRL6*, *SLAMF1*, *APOBEC3H* and the 15q26.1 region with fasting glucose. In silico cross-omics analyses highlight the role of differential methylation in the crosstalk between the adaptive immune system and glucose homeostasis. The differential methylation explains at least 16.9% of the association between obesity and insulin. Our study sheds light on the biological interactions between genetic variants driving differential methylation and gene expression in the early pathogenesis of T2D.

Correspondence and requests for materials should be addressed to J.L. (email: jun.liu@ndph.ox.ac.uk) or to A.D. (email: a.demirkan@surrey.ac.uk) or to C.M.v.D. (email: cornelia.vanduijn@ndph.ox.ac.uk). [#]A full list of authors and their affiliations appears at the end of the paper.

Type 2 diabetes (T2D) is a common metabolic disease, characterized by disturbances in glucose and insulin metabolism. The pathogenesis of T2D is driven by inherited and environmental factors[1]. There is increasing interest in differential DNA methylation in the development of T2D as well as with glucose and insulin metabolism[2–6]. Depending on the region, DNA methylation may result in gene silencing and thus regulate gene expression and subsequent cellular functions[7]. Differential methylation in the circulation may predict the development of future T2D beyond traditional risk factors such as age and obesity[3,8], but it may also be part of the biological mechanism that links age and/or obesity to glucose, insulin metabolism and/or T2D. A recent longitudinal study with multiple visits reported that most DNA methylation changes occur 80–90 days before detectable glucose elevation[9], suggesting that differential DNA methylation evokes changes in glucose and is involved in the early stage(s) of diabetes. Differential DNA methylation is further associated with obesity, which is an important driver of the T2D risk and also precedes the increase in glucose and insulin level in persons developing T2D[8]. A key question to answer is whether the differential methylation associated with glucose and insulin metabolism is an irrelevant epiphenomenon that is related to obesity acting as a statistical confounder or whether there are functional effects of the differential methylation relevant of obesity that is associated to metabolic pathology.

Here, we aim to determine the relation of differential DNA methylation and fasting glucose and insulin metabolism as markers of early stages of diabetes pathology in non-diabetic subjects, accounting for obesity measured as body mass index (BMI). We identify and replicate nine CpG sites associated with fasting glucose (in FCRL6, SLAMF1, APOBEC3H and the 15q26.1 region) and insulin (in LETM1, RBM20, IRS2, MAN2A2 and the 1q25.3 region). Using cross-omics analyses, we present in silico evidence supporting the functional relevance of the CpG sites on the development and progression of diabetes, in terms of their effect on expression paths and elucidate the genetic networks involved.

## Results

**Epigenome-wide association analysis and replication**. In the discovery phase, we performed a blood-based epigenome-wide association study (EWAS) meta-analysis of four cohorts including 4,808 non-diabetic individuals of European ancestry (Supplementary Data 1), which revealed differential DNA methylation at 28 unique CpG sites in either the baseline model without BMI adjustment or in the second model with BMI adjustment (Table 1 and Supplementary Table 1). The summary statistic results of the EWAS are provided as a Data file [https://figshare.com/s/1a1e8ac0fd9a49e2be30]. These include three CpG sites associated with both insulin and glucose, eight CpG sites associated with fasting glucose only and 17 with fasting insulin ($P$ value $< 1.3 \times 10^{-7}$ in meta-analysis). Of these 28 CpG sites, 13 were identified by earlier EWAS studies of either T2D or related traits, including glucose, insulin, hemoglobin A1c (HbA1c), and homeostatic model assessment-insulin resistance (HOMA-IR)[2–5,8,10,11] (Supplementary Table 1). The known CpG sites include three sites located in SLC7A11, CPT1A and SREBF1 that are associated with both glucose and insulin. The remaining ten CpG sites, located in DHCR24, CPT1A, RNF145, ASAM, KDM2B, MYO5C, TMEM49, ABCG1 (harboring two CpG sites) and the 4p15.33 region, are associated with insulin only. All of the previously reported CpG sites with glycemic traits are also associated with BMI in previous EWAS[8,10,12–15] (Supplementary Table 1).

The 15 novel CpG sites were tested using the same statistical models in 11 independent cohorts, including 11,750 non-diabetic participants from the Cohorts for Heart and Aging Research in Genomic Epidemiology (CHARGE) consortium (Supplementary Data 1). Nine unique CpG-trait associations were replicated when correcting for multiple testing using Bonferroni (15 CpGs, $P$ value threshold for significance $< 3.3 \times 10^{-3}$) and were investigated in the further analyses (Table 2). These include five sites (in LETM1, RBM20, IRS2, MAN2A2 and the 1q25.3 region) associated with fasting insulin and one site (in FCRL6) associated with fasting glucose in the baseline model without adjusting for BMI, and three (in SLAMF1, APOBEC3H and the 15q26.1 region, all associated with fasting glucose) emerging in the BMI-adjusted model. Of note, no locus was found to be associated with fasting insulin in the BMI-adjusted model.

Because the replication cohorts also included individuals of African ancestry (AA, $n = 4355$) and Hispanic ancestry (HA, $n = 577$), we also performed the replication stratified by ancestry (Supplementary Data 2). Two CpG sites (cg13222915 and cg18247172) were replicated in the AA population when corrected by the number of tests and two (cg00936728 and cg06229674) replicated with nominal significance. In the HA population, cg20507228 was replicated at nominal significance. Two CpG sites (cg18881723 and cg13222915) show the opposite direction for the effect estimate in HA ancestry population as compared to the other two populations. However, the estimates of effect size are not significantly different from zero ($P$ value $= 0.63$ in cg18881723 and $P$ value $= 0.092$ in cg13222915).

**Glycemic differential DNA methylation and transcriptomics**. To determine whether the differential DNA methylation has functional effects on gene expression and subsequent cellular functions, we conducted three series of analyses. Figure 1 shows the overview of the cross-omics analyses. First, we explored the Genotype-Tissue Expression (GTEx)[16] database for the expression levels of the genes which annotated to the novel CpG sites. We found that the genes are expressed in a wide range of tissues, including whole blood and spleen (in particular MAN2A2 and RBM20), but also other tissues relevant for glucose and insulin metabolism such as adipose subcutaneous, adipose visceral omentum, liver (in particular, SLAMF1, APOBEC3H, FCRL6 and RBM20), pancreas and skeletal muscle (in particular, SLAMF1, APOBEC3H, FCRL6 and MAN2A2) and small intestine terminal ileum (in particular, MAN2A2, RBM20, FCRL6 and APOBEC3H; Supplementary Figure 1).

Second, the effect on gene expression in blood of the previously identified 11 independent CpG sites (cg00574958 in CPT1A and cg06500161 in ABCG1 were used) and the nine novel sites from our current study was examined in the Biobank-based Integrative Omics Study (BIOS) database that is part of the Biobanking and BioMolecular Infrastructure of the Netherlands (BBMRI-NL)[17] (indicated in Fig. 2 in the orange boxes). We found that five CpG sites, i.e. cg00936728 (FCRL6), cg18881723 (SLAMF1), cg00574958 (CPT1A), cg11024682 (SREBF1) and cg06500161 (ABCG1), are expression quantitative trait methylations (eQTMs), i.e. there is correlation between gene expression and methylation[18]. In most cases, the differential methylation levels are associated with the expression (indicated in Fig. 2 in the yellow boxes) of their respective genes. Cg18881723 (SLAMF1) is also associated with the expression of two other genes near SLAMF1, i.e. SLAMF7 and CD244 (Supplementary Table 2).

Third, we investigated whether the genetically regulated expression of the annotated genes in specific tissues is altered in T2D or related traits, such as glucose, insulin and HbA1c. To answer this question, we mined in the MetaXcan database for

**Table 1 CpG sites associated with glycemic traits in discovery phase**

| Locus | CpG | Chr: Pos | Trait | Beta$_{M1}$ | P value$_{M1}$ | Beta$_{M2}$ | P value$_{M2}$ |
|---|---|---|---|---|---|---|---|
| FCRL6 | cg00936728 | 1: 159772194 | Glucose | −1.79 | $9.1 \times 10^{-8‡}$ | −1.60 | $1.9 \times 10^{-7}$ |
| SLAMF1 | cg18881723 | 1: 160616870 | Glucose | 1.16 | $7.5 \times 10^{-8‡}$ | 1.25 | $3.4 \times 10^{-10‡}$ |
| 1q25.3 | cg13222915 | 1: 184598594 | Insulin | −1.69 | $2.6 \times 10^{-9‡}$ | −1.06 | $4.1 \times 10^{-6}$ |
| BRE | cg20657709 | 2: 28509570 | Glucose | −1.42 | $2.7 \times 10^{-6}$ | −1.53 | $4.1 \times 10^{-8‡}$ |
| LRPPRC | cg01913188 | 2: 44223249 | Glucose | 1.18 | $9.4 \times 10^{-6}$ | 1.38 | $5.7 \times 10^{-9‡}$ |
| IRAK2 | cg14527942 | 3: 10276383 | Insulin | 2.44 | $3.4 \times 10^{-10‡}$ | 2.14 | $2.9 \times 10^{-11‡}$ |
| LETM1 | cg13729116 | 4: 1859262 | Insulin | 2.38 | $4.3 \times 10^{-8‡}$ | 1.64 | $4.5 \times 10^{-6}$ |
| RBM20 | cg15880704 | 10: 112546110 | Insulin | 2.50 | $3.8 \times 10^{-9‡}$ | 1.38 | $6.7 \times 10^{-5}$ |
| IRS2 | cg25924746 | 13: 110432935 | Insulin | 2.11 | $3.0 \times 10^{-9‡}$ | 1.32 | $4.9 \times 10^{-6}$ |
| SPTB | cg07119168 | 14: 65225253 | Glucose | −1.64 | $4.4 \times 10^{-7}$ | −1.63 | $4.9 \times 10^{-8‡}$ |
| 15q26.1 | cg18247172 | 15: 91370233 | Glucose | −1.05 | $4.9 \times 10^{-6}$ | −1.18 | $2.8 \times 10^{-8‡}$ |
| MAN2A2 | cg20507228 | 15: 91460071 | Insulin | 1.18 | $5.5 \times 10^{-8‡}$ | 0.87 | $9.0 \times 10^{-7}$ |
| FAM92B | cg06709610 | 16: 85143924 | Insulin | 6.22 | $6.5 \times 10^{-9‡}$ | 6.30 | $5.8 \times 10^{-13‡}$ |
| CD300A | cg08087047 | 17: 72461209 | Glucose | −1.35 | $5.9 \times 10^{-6}$ | −1.45 | $1.1 \times 10^{-7‡}$ |
| APOBEC3H | cg06229674 | 22: 39492189 | Glucose | −1.62 | $1.8 \times 10^{-6}$ | −1.70 | $4.7 \times 10^{-8‡}$ |

Novel epigenome-wide significant results in the discovery phase ($n = 4808$) are shown. Model 1 (M1) indicates inverse variance-weighted fixed effect meta-analysis of effect estimates in four cohorts. Each cohort performed a regression model adjusting for age, sex, technical covariates, white blood cell, and smoking status, and accounting for family structure in family-based cohorts. Model 2 (M2) indicates the meta-analysis of the same studies, adjusting for body mass index (BMI) additionally. Locus: the cytogenetic location or the gene symbol of the CpG sites from Illumina annotation. Beta: effect estimate of the meta-analysis. P value shown is genomic controlled after meta-analysis. The effect refers to the increase/ decrease in fasting glucose/ insulin as the outcome in the model
‡Significant results ($P$ value $< 1.3 \times 10^{-7}$)

**Table 2 CpG sites associated with glycemic traits in replication**

| Locus | CpG | Chr: Pos | Trait | Beta$_{M1}$ | P value$_{M1}$ | Beta$_{M2}$ | P value$_{M2}$ |
|---|---|---|---|---|---|---|---|
| FCRL6 | cg00936728 | 1: 159772194 | Glucose | $-1.55 \times 10^{-3}$ | $9.6 \times 10^{-5‡}$ | NP | NP |
| SLAMF1 | cg18881723 | 1: 160616870 | Glucose | $1.17 \times 10^{-3}$ | $7.7 \times 10^{-3}$ | $1.48 \times 10^{-3}$ | $1.2 \times 10^{-3‡}$ |
| 1q25.3 | cg13222915 | 1: 184598594 | Insulin | $-3.77 \times 10^{-3}$ | $3.3 \times 10^{-16‡}$ | NP | NP |
| BRE | cg20657709 | 2: 28509570 | Glucose | NP | NP | $-9.40 \times 10^{-4}$ | 0.036 |
| LRPPRC | cg01913188 | 2: 44223249 | Glucose | NP | NP | $1.64 \times 10^{-5}$ | 0.90 |
| IRAK2 | cg14527942 | 3: 10276383 | Insulin | $-6.49 \times 10^{-5}$ | 0.48 | $-7.72 \times 10^{-5}$ | 0.45 |
| LETM1 | cg13729116 | 4: 1859262 | Insulin | $1.92 \times 10^{-3}$ | $7.0 \times 10^{-7‡}$ | NP | NP |
| RBM20 | cg15880704 | 10: 112546110 | Insulin | $3.05 \times 10^{-3}$ | $8.6 \times 10^{-12‡}$ | NP | NP |
| IRS2 | cg25924746 | 13: 110432935 | Insulin | $3.38 \times 10^{-3}$ | $3.0 \times 10^{-11‡}$ | NP | NP |
| SPTB | cg07119168 | 14: 65225253 | Glucose | NP | NP | $-7.18 \times 10^{-4}$ | 0.070 |
| 15q26.1 | cg18247172 | 15: 91370233 | Glucose | NP | NP | $-1.77 \times 10^{-3}$ | $5.1 \times 10^{-4‡}$ |
| MAN2A2 | cg20507228 | 15: 91460071 | Insulin | $6.11 \times 10^{-3}$ | $2.3 \times 10^{-15‡}$ | NP | NP |
| FAM92B | cg06709610 | 16: 85143924 | Insulin | $2.08 \times 10^{-5}$ | 0.81 | $5.37 \times 10^{-5}$ | 0.59 |
| CD300A | cg08087047 | 17: 72461209 | Glucose | NP | NP | $-4.92 \times 10^{-4}$ | 0.28 |
| APOBEC3H | cg06229674 | 22: 39492189 | Glucose | NP | NP | $-2.09 \times 10^{-3}$ | $1.4 \times 10^{-6‡}$ |

Novel epigenome-wide significant results in the replication ($n = 11,750$) are shown. Replication was not performed in the non-significant associated model or trait (NP). Model 1 (M1) indicates inverse variance-weighted fixed effect meta-analysis of effect estimates in the 11 cohorts. Each study performed a regression model adjusting for age, sex, technical covariates, white blood cell, and smoking status, and accounting for family structure in family-based cohorts. Model 2 (M2) indicates the meta-analysis of the same studies, adjusting for body mass index (BMI) additionally. Locus: the cytogenetic location or the gene symbol of the CpG sites from Illumina annotation. Beta: effect estimate of the meta-analysis. The effect refers to the increase/ decrease in methylation as the outcome in the model
‡Significant results ($P$ value $< 3.3 \times 10^{-3}$)

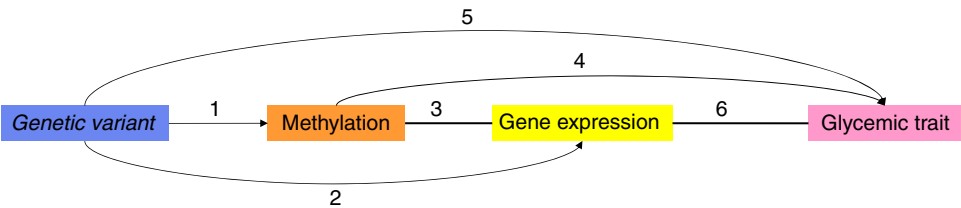

**Fig. 1** Overview of the cross-omics analysis. (1) Methylation quantitative trait loci (meQTL). (2) Expression quantitative trait loci (eQTL). (3) Expression quantitative trait methylation (eQTM). (4) Epigenome-wide association study (EWAS) and Mendelian randomization (MR). (5) Genome-wide association study (GWAS). (6) The association of gene expression expressed in the glucose or insulin metabolism-related tissues and glycemic traits. Results in 1, 2, 3 were extracted from the summary statistics from Biobank-based Integrative Omics Study (BIOS) database ($n = 3814$). Results in 4 was the results in the current EWAS (discovery phase, $n = 4808$, replication phase, $n = 11,750$) and the two-sample Mendelian randomization based on the BIOS database ($n = 3814$) and GWAS results of Meta-Analyses of Glucose and Insulin-related traits Consortium (MAGIC). Results in 5 was from the GWAS results of MAGIC or the DIAbetes Genetics Replication And Meta-analysis consortium (DIAGRAM, $n = 96,496–452,244$). Results in 6 was based on the summary statistics of Genotype-Tissue Expression project (GTEx) and MAGIC or DIAGRAM ($n = 153–491$)

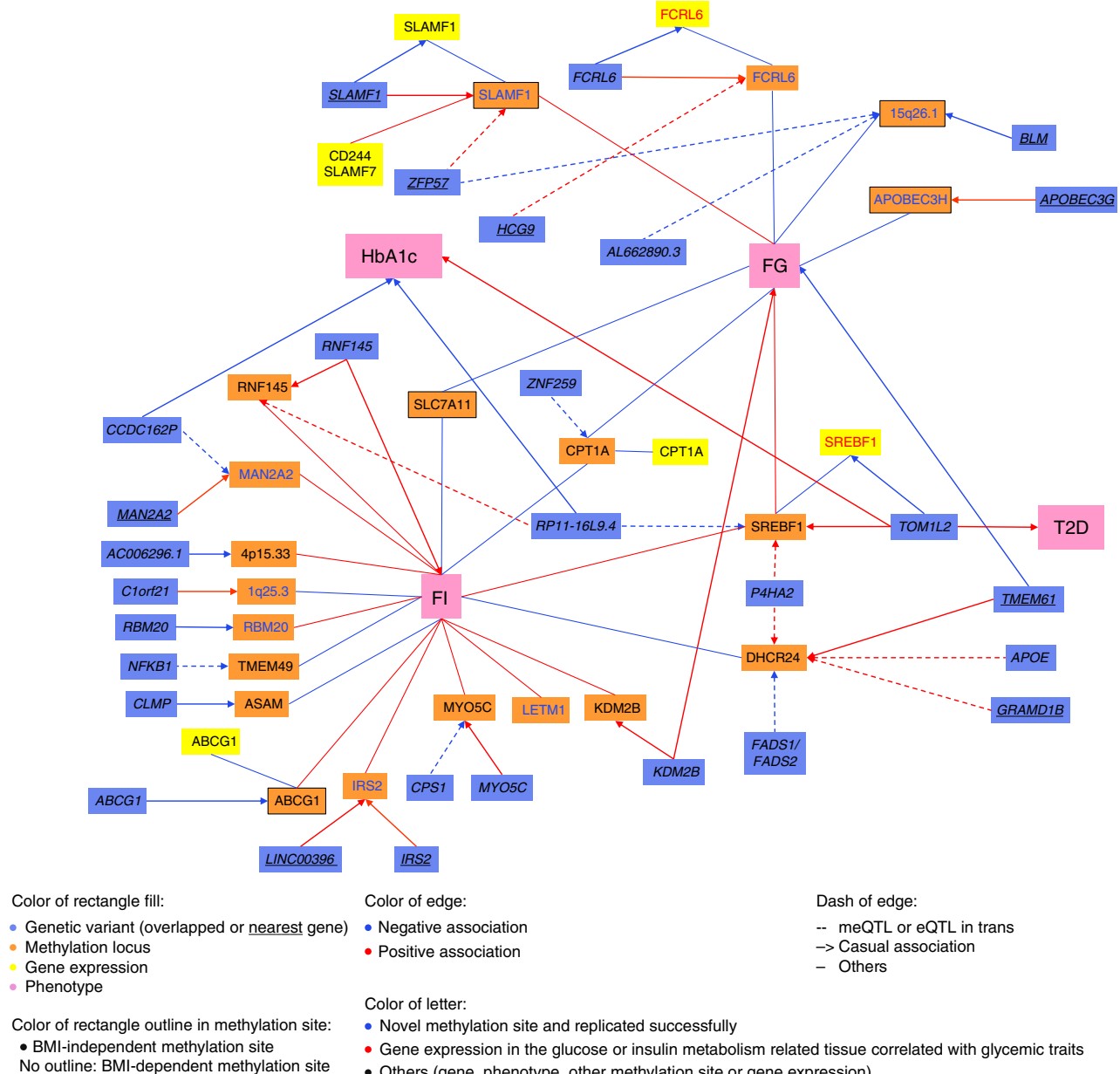

**Fig. 2** Significant associations of the cross-omics integration. The effect allele is standardized across all associations. Only the significant associations which passed the specific *P* value threshold in each association step and the direction of effects consistent were shown in the figure. FG fasting glucose. FI fasting insulin, T2D type 2 diabetes, HbA1c hemoglobin A1c

genome-wide association studies (GWAS) of T2D, fasting glucose, HbA1c, insulin and HOMA-IR[19–23] as a genetic proxy for the traits[24]. No association was found between glycemic traits and the DNA expression in adipose subcutaneous, adipose visceral omentum and small intestine terminal ileum. Supplementary Table 3 gives the significant findings for tissues known to be implicated in glucose and insulin metabolism including blood, liver, pancreas and skeletal muscle (*P* value < 0.05 for MetaXcan). As described earlier, we associated the increased expression of *SREBF1* with decreased risk of T2D and decreased HbA1c levels in the whole blood[25]. The increased expression in the whole blood of *ABOBEC3H*, a methylation locus we identified in the present study, is associated with increased HOMA-IR level, a measure of insulin resistance. In skeletal muscle, the increased fasting glucose is associated with the increased expression of *KDM2B* and decreased expression of *MAN2A2*.

Moreover, we discovered that increased hepatic expression of *FCRL6*, which was annotated to the methylation locus associated with fasting glucose in the present study, is associated with the risk of T2D. In the pancreas, the increased expression of the methylation loci *MYO5C* and *RBM20* are associated with increased fasting glucose levels.

**Glycemic differential DNA methylation and genomics.** Although differential DNA methylation may be the result of environmental exposures, the process is often (partly) heritable with genetic variants (co-)determining the process[26]. Therefore, we next set out to find whether the differential methylation associated with fasting glucose and insulin levels is driven by genetic variants which referred to as methylation quantitative trait loci (meQTLs). Using the BIOS database (blood-based

data)[17], we were able to study 18 out of the 20 unique CpG sites in this respect. We associated 2,991 single-nucleotide polymorphisms (SNPs) in 29 unique meQTLs (indicated in Fig. 2 in the blue boxes) with differential methylation either in *cis* or *trans* (for details see Supplementary Data 3). Six of these meQTLs (4 *cis* and 2 *trans*-acting) are also associated with T2D, fasting glucose, fasting Insulin, or HbA1c in earlier studies[21,22,27–29] and the directions of the effect between the SNP, methylation and glycemic traits are consistent (shown in Fig. 2 in the pink boxes, for details see Supplementary Table 4). A genetic locus near *TMEM61* is a common genetic driver affecting the differential methylation at nearby CpG cg17901584 (*DHCR24*) in our study and fasting glucose levels in an earlier study[22]. Further, the *RNF145* locus was found to be a common driver affecting the differential methylation at cg26403843 (*RNF145*) and fasting insulin levels[21]. The *KDM2B* locus affects differential methylation at cg13708645 (*KDM2B*) and fasting glucose levels[22], and the *TOM1L2/RAI1* locus affects the differential methylation at cg11024682 (*SREBF1*) as well as HbA1c and T2D[27,28]. Two *trans*-acting loci involve a genetic locus in *CCDC162P* that is affecting differential methylation at cg20507228 (*MAN2A2*) and HbA1c[27] and the genetic locus in *RP11-16L9.4* affecting the differential methylation at cg11024682 (*SREBF1*) and HbA1c[27].

We next explored if these genetic variants associated with differential methylation (meQTLs) are also associated with gene expression, i.e. quantitative trait loci (eQTLs; see the integrated outline of analyses in Fig. 2 and detailed in Supplementary Table 5). We searched specifically for expression profiles earlier associated with glycemic CpG sites in blood (listed in Supplementary Table 2). We associated three genetic variants with both differential methylation and gene expression in blood. These include that: 1) rs11265282 in *FCRL6* is positively associated with the differential methylation at cg00936728 (*FCRL6*) and decreased the expression of *FCRL6* in blood, 2) rs1577544 near *SLAMF1* is associated with decreased differential methylation at cg18881723 (*SLAMF1*) and decreased *SLAMF1* expression in blood, and 3) rs6502629 in *TOM1L2* is associated with increased differential methylation at cg11024682 (*SREBF1*) and decreased *SREBF1* expression in blood.

As we observed that the genes driving glycemic CpG sites overlapped with genetic determinants of T2D or related traits, we studied the causal effect of differential methylation on glucose and insulin metabolism with a generalized summary statistic-based Mendelian randomization (MR) test[30]. Up to eight independent genetic variants include in the genetic risk score were used as the instrumental variable for each CpG. Thirteen CpG sites out of the initial 20 met the present MR criteria and were tested by MR (Supplementary Data 4). No significant association was detected when adjusting for multiple testing accounting for 13 independent tests ($P$ value threshold for significance $< 3.8 \times 10^{-3}$). The genetic risk score for cg15880704 (*RBM20*) methylation levels is nominally significantly associated with fasting insulin levels ($P$ value = 0.04), and the genetic risk score for cg18881723 (*SLAMF1*) levels is nominally associated with fasting glucose levels ($P$ value = 0.05) in the MR tests.

**Multi-omics integration and functional annotation**. To understand the biological relevance of our findings, we first integrated the cascade of associations into genomics, epigenomics, transcriptomics and glycemic traits through EWAS, eQTM, meQTL and eQTL. There are three pathways emerging when considering the consistency of the direction of the effects between the associations. One pathway involves *SREBF1*, which in part, was reported earlier[3,25,31] but substantially extended in the current report. The other two involve differential methylation

of *FCRL6* and *SLAMF1* (Fig. 3). The C allele of rs11265282 in *FCRL6* is associated with increased methylation, which turns down the *FCRL6* expression in blood. In addition, the genetically decreased *FCRL6* expression in the liver is also associated with a decreased risk of T2D. The T allele of rs1577544 near *SLAMF1* increases the differential methylation levels in the blood, which decreases *SLAMF1* expression in the circulation, which is consistent with the negative association between the genetic variant and gene expression levels.

To understand the correlation of the findings, we clustered the normalized differential methylation values of the nine novel CpG sites including those not annotated to a gene. Two clusters emerge, one including *IRS2*, *MAN2A2*, 1q25.3 locus (intergenic), *RBM20*, *LETM1* and *SLAMF1* and the second one including *FCRL6*, 15q26.1 (intergenic) and *APOBEC3H* (Fig. 4 and Supplementary Table 6). Four CpGs in *FCRL6*, 15q26.1 (intergenic), *APOBEC3H* and *SLAMF1* are highly correlated with each other, in which the absolute correlation coefficients are bigger than 0.6, while they are located in different chromosomes, suggesting a common biological mechanism: *SLAMF1* and *FCRL6* from chromosome 1, *APOBEC3H* from chromosome 22 and 15q26.1 from chromosome 15. We next performed gene set enrichment analysis in different pathway databases, including KEGG pathways[32], Reactome Pathway Knowledgebase[33] and Gene Ontology (GO) biological process classification[34]. We found that the genes in the first cluster are highly enriched together in multiple pathways, including regulation of leukocyte proliferation, protein secretion and cell activation (*SLAMF1* and *IRS2*), hexose, monosaccharide and carbohydrate metabolism (*IRS2* and *MAN2A2*). Further, *SLAMF1* (cluster 1) and *APOBEC3H* (cluster 2) are both enriched in immune effector processes and innate immune response (Supplementary Table 7).

**BMI in the association of methylation and glycemic traits**. Of note, among the 20 methylation loci associated with glycemic metabolism in the present analyses, 11 are associated with BMI in the previous EWAS[8,10,12–15]. These 11 loci are all associated with insulin metabolism (Supplementary Table 1). Based on the bi-direction MR findings performed as part of the previous EWAS of BMI[8], we found that BMI appears to drive methylation for cg06500161 (*ABCG1*, $P$ value = $6.4 \times 10^{-5}$), a CpG that we associated with insulin levels. Using a marginal $P$ value of 0.05 in their MR results, the differential methylation appears to be a consequence of obesity rather than a cause for three other CpG sites: cg110244682 (*SREBF1*; $P$ value = $4.1 \times 10^{-3}$), cg17901584 (*DHCR24*; $P$ value = $4.1 \times 10^{-3}$) and cg26403843 (*RNF145*; $P$ value = 0.011)[8]. Taken together (Supplementary Table 1), our results raise the question whether BMI is driving differential methylation, which subsequently raises insulin level in the circulation. Such a pathway would predict that the association between BMI and insulin changes when adjusting for differential methylation at *ABCG1*, *SREBF1*, *DHCR24* and *RNF145*. We tested this hypothesis in the non-diabetic individuals of the Rotterdam Study by comparing the relationship between BMI and fasting insulin with and without adjusting for the methylation levels at the four CpG sites. The variance explained ($R^2$) by the linear regression model improves significantly from 0.40 to 0.43 ($P$ value = $1.2 \times 10^{-13}$ by analysis of variance (ANOVA) testing) when adjusting for the CpG effect, while the effect estimates for BMI decrease by 9.2% (beta: 0.065, standard error (SE): 0.003, $P$ value = $1.2 \times 10^{-82}$ for the model without CpG adjustment compared to beta: 0.059, SE: 0.003, $P$ value = $2.9 \times 10^{-70}$ adjusting for the four CpGs). When we extended the adjustment to the 16 CpG sites associated with circulating insulin levels, the variance explained by the model improves further ($R^2 = 0.46$, $P$

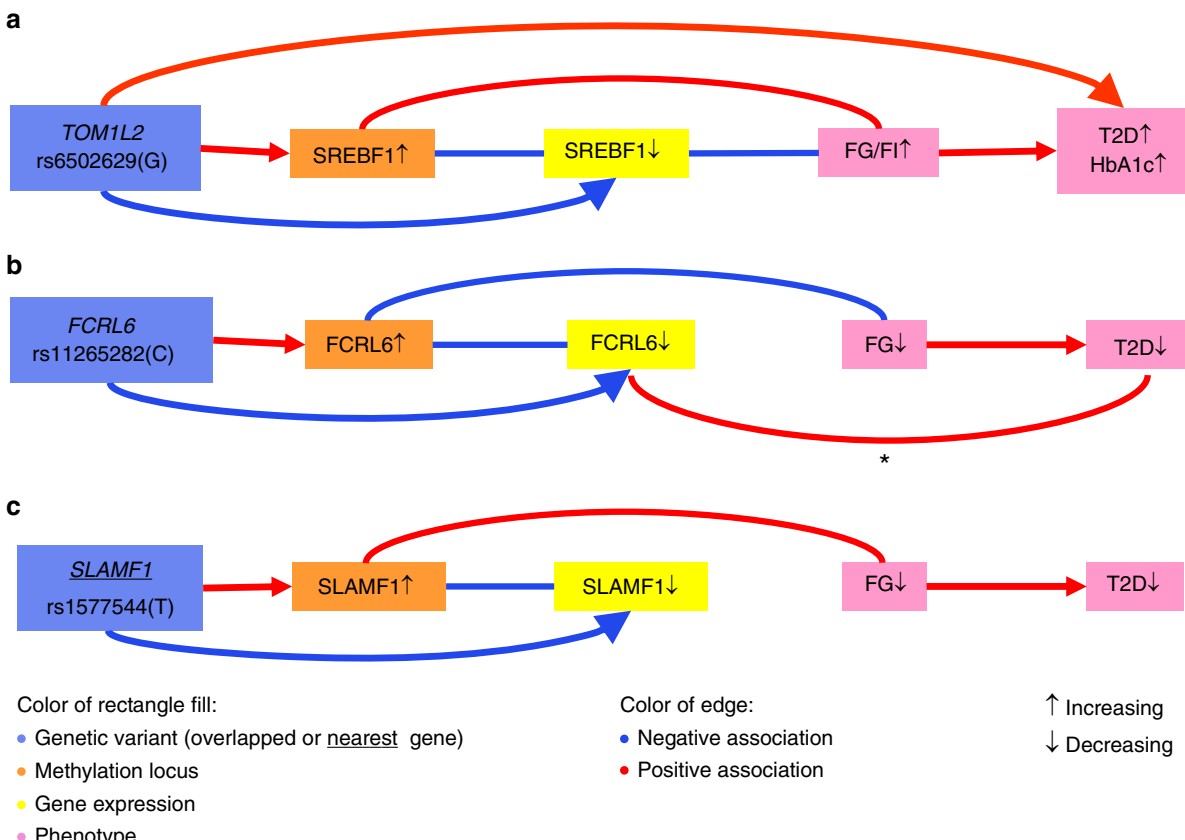

**Fig. 3** The cross-omics integration of CpGs in *SREBF1* (**a**), *FCRL6* (**b**) and *SLAMF1* (**c**). Cascading associations cross multi-omics were integrated in the network. * The association happens in the FCRL6 expression in liver. All other differential methylation or gene expression was measured in blood. FG fasting glucose, FI fasting insulin, T2D type 2 diabetes, HbA1c hemoglobin A1c

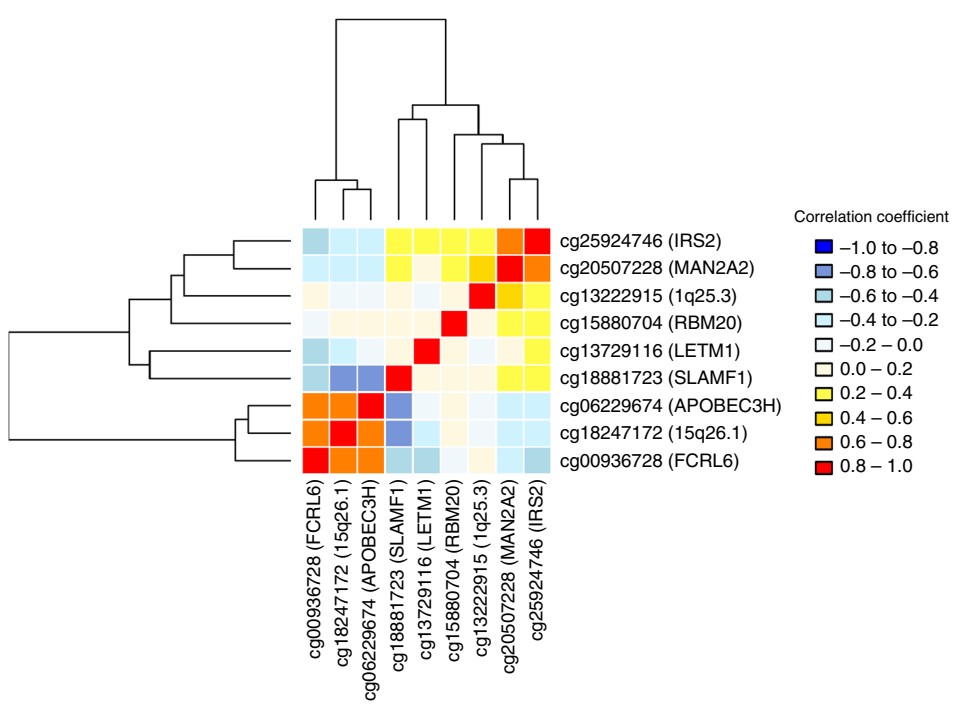

**Fig. 4** Clustered correlation of the nine novel glycemic CpGs. The correlation of the novel CpG sites was checked by Pearson's correlation test ($n = 1544$). The hierarchical cluster analysis was used in the clustering

value $= 2.1 \times 10^{-18}$) and the beta for BMI reduces further by 16.9% (beta: 0.054, SE: 0.003, P value $= 4.6 \times 10^{-58}$ for the model adjusting for 16 CpG sites).

## Discussion

The current large-scale EWAS identify and replicate nine CpG sites associated with fasting glucose (in *FCRL6*, *SLAMF1*, *APOBEC3H* and the 15q26.1 region) or insulin (in *LETM1*, *RBM20*, *IRS2*, *MAN2A2* and the 1q25.3 region). When we adjust for BMI as a potential confounder, three CpG sites (in *SLAMF1*, *APOBEC3H* and the 15q26.1 region) are associated with fasting glucose only after adjustment for BMI. We validate 13 previously reported CpG sites from 11 independent genetic loci[2–6,8,10,12–15] and complement the understanding on why these CpG sites are associated with T2D and/or glycemic traits based on comprehensive cross-omics analyses. We present in silico evidence supporting the functional relevance of the CpG sites, in terms of their effect on expression paths and elucidate the genetic networks involved.

Our data show that differential methylation plays a key role in understanding the immunological changes observed in glucose metabolism[35]. *SLAMF1* and *APOBC3H* are both enriched in immune function and the innate immune response. The differential methylation level at *FCRL6*, 15q26.1 (intergenic), *APOBEC3H* and *SLAMF1* were highly correlated though they were on three different chromosomes. This finding suggests a common pathway. *SLAMF1* belongs to the immunoglobulin gene superfamily and is involved in T-cell stimulation[36]. *APOBEC3H* proteins are part of an intrinsic immune defense that has potent activity against a variety of retroelements[36] and its expression in whole blood is positively associated with HOMA-IR from the current study. *FCRL6* is a distinct indicator of cytotoxic effector T-lymphocytes that is upregulated in diseases characterized by chronic immune stimulation[36]. Meanwhile, we show that decreased *FCRL6* differential methylation increased expression of *FCRL6* and fasting glucose in the blood. A key finding that links *FCRL6* to glucose metabolism is that the genetically determined *FCRL6* expression in the liver is also associated with decreased risk of T2D. In line with a role in immune relation and pathology[37,38], the HLA region (6p22.1 region) is a key meQTLs of *FCRL6* (rs2523946), 15q26.1 (rs3129055 and rs4324798) and *SLAMF1* (rs3129055). Of interest is that in the population of non-diabetic individuals, we found strong signals of the immune system particularly when we adjust the effects attributed to BMI. Remarkably, three out of the four methylation loci at *SLAMF1*, *APOBEC3H* and the 15q26.1 region emerged in the BMI-adjusted model, suggesting that these associations were masked by confounding noise of BMI on methylation in opposite effects to that of insulin.

We studied the interplay between BMI, fasting glucose and insulin levels, and differential methylation in the circulation. On the one hand, we find evidence that the differential methylation of the insulin-related CpG sites together explained up to 16.9% of the association between obesity and insulin levels. These findings are in line with the *Nature* paper on the EWAS of BMI that found that the methylation patterns in blood predict future diabetes[8]. Our study reveals that insulin is a key player underlying the association reported earlier[8]. On the other hand, we find evidence that the association between differential methylation and insulin metabolism is attenuated up to 62%, e.g. CpG sites in *SREBF1* (62%), *ASAM* (56%), *CPT1A* (54%) and *TMEM49* (52%), when BMI is accounted for in the model, suggesting that the interplay between BMI, differential methylation and insulin metabolism is extremely complex and differs across CpG sites. BMI may be a confounder of associations for some CpGs but may be in the causal pathway for others.

To our knowledge, we report for the first time that, in blood, differential methylation of *IRS2* was associated with fasting insulin level. Expression level of *IRS2* (insulin receptor substrate 2) in β-cells in the pancreas are associated with the onset of diabetes[39–41]. Though the expression level of *IRS2* is low in blood, we find its blood-based differential methylation was associated with fasting insulin. We also find an insulin-related genetic locus, *MAN2A2* (mannosidase alpha class 2 A member 2) in our EWAS. *MAN2A2* encodes an enzyme that forms intermediate asparagine-linked carbohydrates (N-glycans)[42]. It is related to the hexose/monosaccharide metabolism. In addition, the expression of *MAN2A2* in skeletal muscle is negatively associated with fasting glucose level and the meQTL (rs9374080) of *MAN2A2* associates with HbA1c[27]. Together, these findings suggest that regulating the differential methylation level or expression level of *MAN2A2* may be relevant to the development of insulin resistance. Another interesting gene that emerged is the familial cardiomyopathy related gene *RBM20*, which may play a role in cardiovascular complications of diabetes via mediating insulin damage in cardiac tissues[43]. The expression of *RBM20* in the pancreas is also associated with fasting glucose. The meQTL for *RBM20* is associated with pulse rate (P value $= 4.6 \times 10^{-5}$) in UKBIOBANK GWAS[44], and its mRNA is highly expressed in cardiac tissues[45].

One limitation of our study is that the main findings are based on data from blood which was the only accessible tissue in our epidemiological studies and may not be representative of more disease-relevant tissues. However, the concordance of differential methylation between blood and adipose is high for certain pathways[46]. DNA methylation globally is considered a relatively stable epigenetic mark that can be inherited through multiple cell divisions[47,48]. However, some changes can be dynamic reflected by recent environmental exposures. This phenomenon could be site-specific. While our study provides a snapshot of associations specific to the fasting state, instant methylation of different CpG sites in the vicinity of *IRS2* and *KDM2B* have been reported earlier[49]. Such effects may also occur at the loci presented in the present study. Our present MR analyses yield no evidence for the causal effects of CpG sites on fasting glucose or insulin. One limitation in the interpretation of the findings is that low power of the MR due to the fact we lack insight in the genes driving differential methylation. For instance, seven of the 13 performed CpG sites have instrumental variables which explain less than 5% of the exposure. Further studies are needed to include additional biologically relevant tissues and perform MR based on the tissue-specific meQLTs. Last but not least, cg19693031 in *TXNIP* has been repeatedly associated with type 2 diabetes case-control status earlier[3,50,51]. Although it did not pass our pre-defined EWAS significance threshold, *TXNIP* is associated with fasting glucose in the non-diabetic population (P value $= 7.6 \times 10^{-7}$ in the BMI adjustment model) if we take the current study aiming to replicate earlier findings. Of note is that cg19693031 is not associated with fasting insulin (in BMI-unadjusted model, p value = 0.30; in the BMI-adjusted model, p value = 0.37).

In conclusion, our large-scale EWAS and replication identifies nine differentially methylated sites associated with fasting glucose or insulin, and shows that differential methylation explains part of the association between obesity and insulin metabolism. The integrative in silico cross-omics analyses provide insights of glycemic loci into the genetics, epigenetics and transcriptomics pathways. We also highlight that differential methylation is a key point in the involvement of the adaptive immune system in glucose homeostasis. Further studies in the future will benefit from tissue-specific methylation and meQTL databases which are currently the missing piece of the in silico data integration framework.

## Methods

**Study population**. The discovery samples consisted of 4808 European individuals without diabetes from four non-overlapped cohorts, recruited by Rotterdam Study III-1 (RS III-1, $n = 626$), Rotterdam Study II-3 and Rotterdam Study III-2 (called as RS-BIOS, $n = 705$), Netherlands Twin Register (NTR, $n = 2753$) and UK adult Twin registry (TwinsUK, $n = 724$). The replication sets contained up to 11,750 individuals from 11 independent cohorts from CHARGE, including up to 6818 individuals from European ancestry, 4355 from African ancestry and 577 from Hispanic ancestry (Supplementary Data 1). They are from Atherosclerosis Risk in Communities (ARIC) Study, Baltimore Longitudinal Study of Aging (BLSA), Cardiovascular Health Study (CHS), Framingham Heart Study Cohort (FHS), The Genetic Epidemiology Network of Arteriopathy (GENOA), Genetics of Lipid Lowering Drugs and Diet Network (GOLDN), Hypertension Genetic Epidemiology Network (HyperGEN), Invecchiare in Chianti Study (InCHIANTI), Kooperative Gesundheitsforschung in der Region Augsburg (KORA), Women's Health Initiative - Broad Agency Award 23 (WHI-BAA23) and Women's Health Initiative - Epigenetic Mechanisms of PM-Mediated CVD (WHI-EMPC). We excluded individuals with known diabetes and/or fasting glucose ≥ 7 mmol/l and/or those on anti-diabetic treatment. All studies were approved by their respective Institutional Review Boards, and all participants provided written informed consent. Details about the studies have been reported previously, and the key references as well as the summary of the design of each study are reported in Supplementary Note 1.

**Glycemic traits and covariates**. Venous blood samples were obtained after an overnight fast in all discovery and replication cohorts. BMI was calculated as weight over height squared (kg m$^{-2}$) based on clinical examinations. Smoking status was divided into current, former and never, based on questionnaires. White blood cell counts were quantified using standard laboratory techniques or predicted from methylation data using the Houseman method[52]. The cohort-specific measurement of glycemic traits and covariates are shown in Supplementary Note 1.

**DNA methylation quantification**. The Illumina© Human Methylation450 array was used in all discovery and replication cohorts to quantify genome-wide DNA methylation in blood samples. We obtained DNA methylation levels reported as $\beta$ values, which represents the cellular average methylation level ranging from 0 (fully unmethylated) to 1 (fully methylated). Study-specific details regarding DNA methylation quantification, normalization and quality control procedures are provided in the Supplementary Note 1.

**Epigenome-wide association analysis and replication**. All statistical analyses were performed using $R$ statistical software and the two-tailed test was considered. Insulin was natural log transformed. In the discovery analysis, we first performed EWAS in each cohort separately. Linear regression analysis was used to test the association between glucose and insulin with each CpG site in the Rotterdam Study samples. Linear mixed models were used in NTR and TwinsUK, accounting for the family structure. We fitted the following two models for each cohort: (1) the baseline model adjusting for age, sex, technical covariates (chip array number and position on the array), white blood cell counts (lymphocytes, monocytes, and granulocytes) and smoking status, and (2) a second model additionally adjusting for BMI. We removed probes that have evidence of multiple mapping or contain a genetic variant in the CpG site[53]. All cohort-specific EWAS results for each model were then meta-analysed using inverse variance-weighted fixed effect meta-analysis as implemented in the metafor R package[54]. In total, we meta-analysed 393,183 CpG sites that passed quality control in all four discovery cohorts. The details of the quality control for each cohort could be found in the Supplementary Note 1. The association was later corrected by the genomic control factor ($\lambda$) in each meta-EWAS[55]. We produced quantile-quantile (QQ) plots of the -log$_{10}$ ($P$) to evaluate inflation in the test statistic (Supplementary Figure 2). A Bonferroni correction was used to correct for multiple testing and identify epigenome-wide significant results ($P < 1.3 \times 10^{-7}$). We did not correct the number of glycemic traits and models, as they are highly correlated and not independent. The genome coordinates were provided by Illumina (GRCh37/hg19). The CpG sites were annotated to genes using Infinium HumanMethylation 450 BeadChip annotation resources. The correlation of the CpG sites located in the same gene was further checked in the overall RS III-1 and RS-BIOS samples by Pearson's correlation test ($n = 1544$) to find the independent top CpG sites.

For the associations discovered in the meta-EWAS that have not been reported previously, we attempted replication in independent samples using the same traits and regression models as in the discovery analyses. Study-specific details of replication cohorts are provided in Supplementary Data 1 and Supplementary Note 1. Results from each replication cohort were meta-analysed using the same methods as in the discovery analyses. Bonferroni $P$ value $< 3.3 \times 10^{-3}$ (0.05 corrected by 15 CpGs tested for associations) was considered significant.

**Glycemic differential DNA methylation and transcriptomics**. To explore whether the differential CpG sites were associated with gene expression level in blood, we explored eQTMs[17] from the European blood-based BIOS database[17] from BBMRI-NL which captured meQTLs, eQTLs and eQTMs from genome-wide database of 3841 Dutch blood samples (See resources of the database in URLs). The

associated gene expression probes of the known and replicated CpG sites were searched. We then tested whether the expression of the genes that harbor the identified methylation sites was associated with T2D and related traits in glucose metabolism-related tissues (adipose subcutaneous, adipose visceral omentum, liver, whole blood, pancreas, skeletal muscle and small intestine terminal ileum) using MetaXcan package[24,56]. MetaXcan associates the expression of the genes with the phenotype by integrating functional data generated by large-scale efforts, e.g. GTEx project[16] with that of the GWAS of the trait. MetaXcan is trained on transcriptome models in 44 human tissues from GTEx and is able to estimate their tissue-specific effect on phenotypes from GWAS. For this study, we used the GWAS studies of T2D[19], fasting glucose traits[21,22], fasting insulin[22], HbA1c[23] and HOMA-IR[20]. We used the nominal $P$ value threshold ($P$ value threshold for significance < 0.05) as we had separate assumptions for each terminal pathway between gene expressions and phenotype. The associations with genes in low prediction performance were excluded, i.e. the association of the tissue model's correlation to the gene's measured transcriptome is not significant ($P$ value > 0.05).

**Glycemic differential DNA methylation and genomics**. We identified the genetic determinants of the significant CpG sites known or replicated through the current EWAS using the results of the *cis* and *trans* meQTLs from the European blood-based BIOS database[17] (See resources of the database in URLs). All the reported SNPs with $P$ value adjusted for false discovery rate (FDR) less than 0.05 in the database were treated as the target genetic variants in the present study. The SNPs were annotated based on the information in the BIOS study[17] or the nearest protein-coding gene list from SNPnexus[57] on GRCh37/hg19. We also explored the associations of these DNA methylation-related genetic variants with T2D or related traits, i.e., fasting glucose, insulin, HbA1c and HOMA-IR, based on public GWAS data sets in European ancestry[20–22,27–29]. Meanwhile, we checked the effect direction consistency of the association between the SNPs, CpG sites and T2D or related traits. That is the direction of the association between SNP and T2D or related traits should be a combination of the direction of SNP with CpG sites and CpG sites with T2D or related traits. A multiple-testing correction was performed by Bonferroni adjustment ($P$ value significant threshold < 1.8 × 10$^{-3}$, 0.05 corrected by the 29 genetic loci shown in Supplementary Data 3). The associations of the DNA methylation-related genetic variants and the gene expression were also looked up in the BIOS database[17]. This is limited to the expression profiles earlier associated with glycemic CpG sites in blood.

For the significant CpG sites known or replicated through EWAS, we attempted to evaluate the causality effect of CpG sites on their significant traits, either fasting glucose or fasting insulin, using two-sample MR approach as described in detail before by Dastani et al.[30,58] based on the summary statistic GWAS results from the BIOS database and the Meta-Analyses of Glucose and Insulin-related traits Consortium (MAGIC) database[17,21] (Supplementary Figure 3). Briefly, we constructed a weighted genetic risk score for individual CpG on phenotype using independent SNPs as the instrument variables of the CpG, implemented in the R-package gtx. The effect of each score on phenotype was calculated as

$$\text{ahat} = \frac{\sum(\omega_i \beta_i / s_i^2)}{\sum(\omega_i^2 / s_i^2)},$$

where $\beta_i$ is the effect of the CpG-increasing alleles on phenotype, $s_i$ its corresponding standard error and $\omega_i$ the SNP effect on the respective CpG. Because the genetic variants might be close (*cis*) or far (*trans*) from the methylated site, we also performed MR test in the *cis* only SNPs if the CpG has both *cis* and *trans* genetic markers. All SNPs were mapped to the human genome build hg19. For each test (one CpG with one trait), we extracted all the genetic markers of the CpG in the fasting glucose or insulin GWAS from the MAGIC data set ($n = 96,496$)[21] with their effect estimate and standard error on fasting glucose or insulin. Within the overlapped SNPs, we removed SNPs in potential linkage disequilibrium (LD, pairwise $R^2 \geq 0.05$) in 1-Mbp window based on the 1000 Genome imputed genotype data set from the general population: Rotterdam Study I (RS I, $n = 6291$)[59]. We managed to exclude the genetic loci which were genome-wide associated with glycemic traits, but none of the genetic loci meet this exclusion criterion. The instrumental variables that explain more than 1% of the variance in exposure (DNA methylation) were taken forward for MR test. The Bonferroni $P$ value threshold was used to correct for the 13 CpG sites available for MR ($P$ value < 3.8 × 10$^{-3}$).

**Functional annotation**. Further, we integrated the cascade of associations as above among the results of EWAS, eQTM, meQTL and eQTL and showed in Fig. 3. We checked the effect direction consistency of the association between the SNPs, CpG sites, gene expression in blood and glycemic traits. The correlation of the novel CpG sites was checked in the overall RS III-1 and RS-BIOS samples by Pearson's correlation test ($n = 1544$). The hierarchical cluster analysis was used in the clustering. Gene set enrichment analyses were performed in the genes of new CpG sites[60]. We tested if genes of interest were over-represented in any of the predefined gene sets from KEGG pathway database[32], Reactome Pathway Knowledgebase[33] and GO biological process[34]. Multiple test correction was performed in the tests. Gene sets of KEGG pathway database, Reactome Pathway Knowledgebase were obtained from Molecular Signatures Database (MsigDB) c2 and GO biological process was obtained from MsigDB c5[60]. We used the platform of Functional

Mapping and Annotation of Genome-wide Association Studies (FUMA GWAS)[61] and GENE2FUNC function to perform the gene set enrichment analysis and the tissue-specific gene expression patterns based on GTEx v6[16]. Besides, the tools Ensembl Human Genes[62] (see URLs) and UCSC GRCh37/hg19[63] (see URLs) were also used in interpreting genetic determinants, CpG sites and genes.

**BMI in the association of methylation and glycemic traits**. We used linear regression to check the effect of CpGs on the relationship between BMI and fasting insulin in the non-diabetic individuals in Rotterdam study. The initial model used BMI as the independent variable and the natural log transformed insulin as the dependent variable. The covariates included age, sex, technical covariates (chip array number and position on the array), white blood cell counts, smoking status and data set (RS III-1 and RS-BIOS). The normalized differential methylation values of CpG sites were added as covariates in the advanced model. The differences of the models were compared by ANOVA testing using anova function in R ($P$ value < 0.05).

**URLs**. BIOS database, https://genenetwork.nl/biosqtlbrowser/ [https://genenetwork.nl/biosqtlbrowser/]; SNPnexus, http://snp-nexus.org/index.html [http://snp-nexus.org/index.html]; GWAS database of glycemic traits, https://www.magicinvestigators.org/ [https://www.magicinvestigators.org/]; GWAS database of T2D, http://diagram-consortium.org/ [http://diagram-consortium.org/]; MetaXcan, https://s3.amazonaws.com/imlab-open/Data/MetaXcan/results/metaxcan_results_database_v0.1.tar.gz [https://s3.amazonaws.com/imlab-open/Data/MetaXcan/results/metaxcan_results_database_v0.1.tar.gz]; NHGRI-EBI Catalog, https://www.ebi.ac.uk/gwas/ [https://www.ebi.ac.uk/gwas/]; Ensembl, https://www.ensembl.org/Homo_sapiens/Info/Index [https://www.ensembl.org/Homo_sapiens/Info/Index]; FUMA, http://fuma.ctglab.nl [http://fuma.ctglab.nl]; UCSC, https://genome.ucsc.edu/cgi-bin/hgGateway [https://genome.ucsc.edu/cgi-bin/hgGateway] (available: 1st Jan 2019)

**Reporting Summary**. Further information on research design is available in the Nature Research Reporting Summary linked to this article.

## Data Availability

All relevant data supporting the key findings of this study are available within the article and its Supplementary Information files; the cohort data sets generated and analyzed during the current study are available from the authors from each cohort upon reasonable request. The summary statistics of each cohort and meta-analysis in the discovery phase and the source data underlying Supplementary Figure 2 are provided as a Data file [https://figshare.com/s/1a1e8ac0fd9a49e2be30]. The web links for the publicly available data sets used in the paper are listed in URLs. In detail, for the BIOS data, the cis-meQTL look-up files were mainly from "Full list of primary cis-meQTLs" and the results in "Cis-meQTLs independent top effects" were also checked. The trans-meQTL look-up file was from "Trans-meQTLs top effects". The "eQTM" look-up file was from "Cis-eQTMs independent top effects". The eQTL look-up file was from "Cis-eQTLs Gene-level all primary effects". Fasting glucose GWAS was from both ftp://ftp.sanger.ac.uk/pub/magic/MAGIC_Metabochip_Public_data_release_25Jan.zip [ftp://ftp.sanger.ac.uk/pub/magic/MAGIC_Metabochip_Public_data_release_25Jan.zip] and ftp://ftp.sanger.ac.uk/pub/magic/MAGIC_Manning_et_al_FastingGlucose_MainEffect.txt.gz [ftp://ftp.sanger.ac.uk/pub/magic/MAGIC_Manning_et_al_FastingGlucose_MainEffect.txt.gz]; fasting insulin GWAS was from ftp://ftp.sanger.ac.uk/pub/magic/MAGIC_Manning_et_al_lnFastingInsulin_MainEffect.txt.gz [ftp://ftp.sanger.ac.uk/pub/magic/MAGIC_Manning_et_al_lnFastingInsulin_MainEffect.txt.gz]; HbA1c was from ftp://ftp.sanger.ac.uk/pub/magic/HbA1c_METAL_European.txt.gz [ftp://ftp.sanger.ac.uk/pub/magic/HbA1c_METAL_European.txt.gz]. The type 2 diabetes GWAS was downloaded from http://diagram-consortium.org/ [http://diagram-consortium.org/] "T2D GWAS meta-analysis - Trans-Ethnic Summary Statistics Published in in Mahajan et al. (2018)". The file "T2D_TranEthnic.BMIunadjusted.txt" was used. A reporting summary for this article is available as a supplementary information file.

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

## Acknowledgements

We gratefully acknowledge the BIOS consortium (https://www.bbmri.nl/?p=259) of Biobanking and BioMolecular resources Research Infrastructure of the Netherlands (BBMRI-NL) and Cohorts for Heart and Aging Research in Genomic Epidemiology (CHARGE) consortium. This work is part of the CardioVasculair Onderzoek Nederland (CVON 2012-03), the Common mechanisms and pathways in Stroke and Alzheimer's disease (CoSTREAM) project (https://www.costream.eu, grant agreement No 667375), Memorabel program (project number 733050814), Netherlands X-omics Research Infrastructure and U01-AG061359 NIA. The full list of funding information of each cohort is found in Supplementary Note 2. J.L., C.M.v.D. and A.Demirkan have used exchange grants from the Personalized pREvention of Chronic DIseases consortium (PRECeDI) (H2020-MSCA-RISE-2014). A.Demirkan is supported by a Veni grant (2015) from ZonMw (VENI 91616165). C.M.v.D. received funding of CardioVasculair Onderzoek Nederland (CVON2012-03) of the Netherlands Heart Foundation. B.A.H. was supported by NHLBI K01 award (K01 HL130609-02). V.W.V.J. received a grant from the Netherlands Organization for Health Research and Development (VIDI 016.136.361) and a Consolidator Grant from the European Research Council (ERC-2014-CoG-648916). J.F.F. has received funding from the European Union's Horizon 2020 research and innovation programme under grant agreement No 633595 (Dyna-HEALTH). J.B.M. is supported by K24 DK080140. J.T.B. received funding support from the JPI ERA-HDHL DIMENSION project (BBSRC BB/S020845/1) and from the ESRC (ES/N000404/1). The views expressed in this manuscript are those of the authors and do not necessarily represent the views of the National Heart, Lung, and Blood Institute; the National Institutes of Health; or the U.S. Department of Health and Human Services.

## Author contributions

J.L., E.C.-M., D.L., A.Dehghan, J.T.B., J.B.J.v.M., A.I., A. Demirkan and C.M.v.D. contributed to study design. J.L., S. Ligthart, P.R.M., M.J.P., L.D., V.W.V.J., J.F.F., G.W., E.J. C.d.G, T.L.A., L.H., E.A.W., N.A., T.D.S., M.-F.H., J.I.R., J.B.J.v.M., A.I., D.I.B., J.T.B. and C.M.v.D. contributed to data collection. V.W.V.J., H.T., G.W., E.J.C.d.G., D.L., J.A.S., N.S., S.B., L.F., D.K.A., H.G., T.L.A., L.H., A.B., E.A.W., A.G.U., E.J.G.S., O.H.F., J.D., J.I. R., J.B.M., J.P., D.I.B., T.D.S. and C.M.v.D. contributed to cohort design and management. J.L., E.C-M., J.v.D., S.Lent, I.N., P.-C.T., T.C.M., R.J., J.F.F., A.Y.C., S.-J.H., R.G., E.S., C.H., B.A.H., T.T., A.Z.M., R.N.L., M.A.J. and A.I. contributed to data analysis. J.L., E.C.-M., S.Ligthart, K.W.v.D., N.A., A.Dehghan, A.I., A. Demirkan and C.M.v.D. contributed to writing of manuscript. J.L., E.C.-M., J.v.D., S. Lent, S. Ligthart, T.C.M., L.D., V.W.V.J., H.T., J.F.F., D.L., J.B., R.G., C.H., B.A.H., N.S., D.K.A., H.G., E.A.W., N.A., E.J. G.S., J.D., M.-F.H., J.I.R., J.B.M., J.P., A.I., J.T.B., A. Demirkan and C.M.v.D. contributed to critical review of manuscript.

## Additional information

**Competing interests:** The authors declare no competing interests.

Jun Liu [1,2,54], Elena Carnero-Montoro [1,3,4,54], Jenny van Dongen [5], Samantha Lent [6], Ivana Nedeljkovic [1], Symen Ligthart [1], Pei-Chien Tsai [4,7,8], Tiphaine C. Martin [4,9,10], Pooja R. Mandaviya [11], Rick Jansen [12], Marjolein J. Peters [11], Liesbeth Duijts [13,14], Vincent W.V. Jaddoe [1,15,16], Henning Tiemeier [17,18], Janine F. Felix [1,15,16], Gonneke Willemsen [5], Eco J.C. de Geus [5], Audrey Y. Chu [19,20], Daniel Levy [19,20], Shih-Jen Hwang [19,20], Jan Bressler [21], Rahul Gondalia [22], Elias L. Salfati [23], Christian Herder [24,25,26], Bertha A. Hidalgo [27], Toshiko Tanaka [28], Ann Zenobia Moore [28], Rozenn N. Lemaitre [29], Min A Jhun [30], Jennifer A. Smith [30], Nona Sotoodehnia [29], Stefania Bandinelli [31], Luigi Ferrucci [28], Donna K. Arnett [32], Harald Grallert [24,33], Themistocles L. Assimes [23], Lifang Hou [34,35], Andrea Baccarelli [36], Eric A. Whitsel [22,37], Ko Willems van Dijk [38,39], Najaf Amin [1], André G. Uitterlinden [1,11], Eric J.G. Sijbrands [11], Oscar H. Franco [1,40], Abbas Dehghan [1,41], Tim D. Spector [4], Josée Dupuis [6], Marie-France Hivert [42,43,44], Jerome I. Rotter [45], James B. Meigs [46,47,48], James S. Pankow [49], Joyce B.J. van Meurs [50], Aaron Isaacs [1,50], Dorret I. Boomsma [5], Jordana T. Bell [4], Ayşe Demirkan [1,51,52,55] & Cornelia M. van Duijn [1,2,53,55]

[1]Department of Epidemiology, Erasmus University Medical Center, Rotterdam 3015GD, The Netherlands. [2]Nuffield Department of Population Health, University of Oxford, Oxford OX3 7FL, UK. [3]Center for Genomics and Oncological Research, GENYO, Pfizer/University of Granada/Andalusian Government, PTS, Granada 18007, Spain. [4]Department of Twin Research and Genetic Epidemiology, King's College London, London WC2R 2LS, UK. [5]Department of Biological Psychology, Amsterdam Public Health (APH) research institute, Amsterdam UMC, Vrije Universiteit Amsterdam, Amsterdam 1081BT, The Netherlands. [6]Department of Biostatistics, Boston University School of Public Health, Boston, MA 02118, USA. [7]Department of Biomedical Sciences, Chang Gung University, Taoyuan 333, Taiwan. [8]Division of Allergy, Asthma, and Rheumatology, Department of Pediatrics, Chang Gung Memorial Hospital, Linkou 333, Taiwan. [9]Department of Oncological Sciences, Icahn School of Medicine at Mount Sinai, New York, NY 10029, USA. [10]The Tisch Cancer Institute, Icahn School of Medicine at Mount Sinai, New York, NY 10029, USA. [11]Department of Internal Medicine, Section of Pharmacology Vascular and Metabolic Diseases, Erasmus University Medical Center, Rotterdam 3015GD, The Netherlands. [12]Department of Psychiatry and Amsterdam Neuroscience, Amsterdam UMC, Vrije Universiteit Amsterdam, Amsterdam 1081BT, The Netherlands. [13]Division of Neonatology, Department of Pediatrics, Erasmus University Medical Center, Rotterdam 3015GD, The Netherlands. [14]Division of Respiratory Medicine, Department of Pediatrics, Erasmus University Medical Center, Rotterdam 3015GD, The Netherlands. [15]Department of Pediatrics, Erasmus University Medical Center, Rotterdam 3015GD, The Netherlands. [16]Generation R Study Group, Erasmus University Medical Center, Rotterdam 3015GD, The Netherlands. [17]Department of Child and Adolescent Psychiatry, Erasmus University Medical Center, Rotterdam 3015GD, The Netherlands. [18]Department of Social and Behavioral Science, Harvard TH Chan School of Public Health, Boston, MA 02115, USA. [19]The Population Sciences Branch, Division of Intramural Research, National Heart, Lung and Blood Institute, National Institutes of Health, Bethesda, MD 20814, USA. [20]The Framingham Heart Study, National Heart, Lung and Blood Institute, National Institutes of Health, Framingham, MA 01702, USA. [21]Human Genetics Center, School of Public Health, University of Texas Health Science Center at Houston, Houston, TX 77030, USA. [22]Department of Epidemiology, Gillings School of Global Public Health, University of North Carolina, Chapel Hill, NC 27599, USA. [23]Department of Medicine, Stanford University School of Medicine, Stanford, CA 94305, USA. [24]German Center for Diabetes Research (DZD), München-Neuherberg 85764, Germany. [25]Institute for Clinical Diabetology, German Diabetes Center, Leibniz Center for Diabetes Research at Heinrich Heine University Düsseldorf, Düsseldorf 40225, Germany. [26]Division of Endocrinology and Diabetology, Medical Faculty, Heinrich Heine University Düsseldorf, Düsseldorf 40225, Germany. [27]Department of Epidemiology, University of Alabama at Birmingham, Birmingham, AL 35233, USA. [28]Translational Gerontology Branch, National Institute on Aging, Baltimore, MD 21224, USA. [29]Cardiovascular Health Research Unit, Department of Medicine, University of Washington, Seattle, WA 98101, USA. [30]Department of Epidemiology, School of Public Health, University of Michigan, Ann Arbor, MI 48109, USA. [31]Geriatric Unit, Azienda Sanitaria di Firenze, Florence 50137, Italy. [32]School of Public Health, University of Kentucky, Lexington, KY 40536, USA. [33]Research Unit of Molecular Epidemiology, Institute of Epidemiology, Helmholtz Zentrum München Research Center for Environmental Health, Neuherberg 85764, Germany. [34]Center for Population Epigenetics, Robert H. Lurie Comprehensive Cancer Center, Feinberg School of Medicine, Northwestern University Chicago, Evanston, IL 60611, USA. [35]Department of Preventive Medicine, Feinberg School of Medicine, Northwestern University, Chicago, IL 60611, USA. [36]Department of Environmental Health Sciences, Mailman School of Public Health, Columbia University, New York, NY 10032, USA. [37]Department of Medicine, University of North Carolina School of Medicine, Chapel Hill, North Carolina, NC 27516, USA. [38]Department of Human Genetics, Leiden University Medical Center, Leiden 2333ZA, The Netherlands. [39]Department of Medicine, Division of Endocrinology, Leiden University Medical Center, Leiden 2333ZA, The Netherlands. [40]Institute of Social and Preventive Medicine (ISPM), University of Bern, Bern 3012, Switzerland. [41]Department of Epidemiology and Biostatistics, Imperial College London, London SW7 2AZ, UK. [42]Department of Medicine, Université de Sherbrooke, Sherbrooke, QC J1K0A5, Canada. [43]Diabetes Unit, Massachusetts General Hospital, Boston, MA 02114, USA. [44]Division of Chronic Disease Research Across the Lifecourse, Department of Population Medicine, Harvard Medical School and Harvard Pilgrim Health Care Institute, Boston, MA 02215, USA. [45]The Institute for Translational Genomics and Population Sciences and Departments of Pediatrics and Medicine, Los Angeles Biomedical Research Institute at Harbor-UCLA Medical Center, Torrance, CA 90502, USA. [46]Department of Medicine, Harvard Medical School, Boston, MA 02115, USA. [47]Division of General Internal Medicine, Massachusetts General Hospital, Boston, MA 02114, USA. [48]Programs in Metabolism and Medical & Population Genetics, Broad Institute of MIT and Harvard, Cambridge, MA 02142, USA. [49]Division of Epidemiology and Community Health, School of Public Health, University of Minnesota, Minneapolis, MN 55455, USA. [50]CARIM School for Cardiovascular Diseases, Maastricht Centre for Systems Biology (MaCSBio), and Departments of Biochemistry and Physiology, Maastricht University, Maastricht 6211LK, The Netherlands. [51]Department of Genetics, University Medical Center Groningen, Groningen 9713GZ, The Netherlands. [52]Section of Statistical Multi-Omics, Department of Experimental and Clinical Research, School of Bioscience and Medicine, Univeristy of Surrey, Guildford GU2 7XH, UK. [53]Leiden Academic Center for Drug Research, Leiden University, Leiden 2311EZ, The Netherlands. [54]These authors contributed equally: Jun Liu, Elena Carnero-Montoro. [55]These authors jointly supervised this work: Ayşe Demirkan, Cornelia M. van Duijn

