## [Peer Review File · Nature Communications]

Reviewers' comments:

Reviewer #1 (Remarks to the Author):

This is a large scale study which maps DNA methylation changes associated with fasting glucose and insulin in non-diabetic individuals. The study is based on the assumption that genetic variants drive methylation status of CpGs, which in turn regulate gene expression. Like in numerous other EWAS, only a small number of statistically significant CpG methylation differences were detected, and their functional roles are not clear.

1. The study is carefully performed and followed the well-known rules for epigenome wide association studies, including mendelian randomization, and does not have any evident major limitations or shortcomings. The main problem is that it lacks innovation and new insight. Pages are full of gene names, p values, and other technical details which do not lead into any new understanding of the role of epigenetic factors in the phenotypes of interest. Only researchers working on glucose metabolism related traits and diseases will benefit from this paper.
2. One evident and important question stemming from the study design, but not addressed/discussed by the authors, is the relationship between dynamic levels of glucose/insulin and cytosine methylation which assumed to be static in terminally differentiated somatic cells. The dozen or so cytosines detected under "fasting" conditions – would they also show an association with glucose/insulin after a meal? Or it will be a different set cytosines whose methylation status correlate with high(er) levels of glucose/insulin?
3. All findings are provided by Z scores and p values, which are not informative in terms of the actual DNA methylation differences and effect sizes. Since the final CpG set is quite small it would be feasible to show their profiles (i.e. scatterplot of correlation with both fasting insulin and glucose). Or to present the coefficients from the related models in an interpretable form (i.e. increase/decrease in fasting insulin per 1% of methylation). This would enhance the interpretability of results, enable comparisons between CpGs and their effects on insulin and glucose as well as provide insight into biological meaningfulness of the effect.
4. It is not clear from the Methods if Houseman algorithm was used for correcting cell counts in the African population which was quite substantial (n=4355). The question is if the algorithm has been validated for non-Caucasian populations. Is it possible this was the reason that findings in European populations were not replicated in other ethnic groups (lines 165-168). "Consistent directions" is not a replication.
5. Bonferroni correction was used to adjust the p-values based on the number of CpGs but not applied for the number of models because the models are highly correlated. Are their correlations stable and do not change depending on age, BMI, some other parameters? E.g. how about insulin and glucose correlations in individuals with high BMI and insulin resistance?
6. Editing is required. E.g. line 197 "were testes" – were tested; line 319 Linear mixed models were used in NTR and TwinsUK accounting for the family structure.

Reviewer #2 (Remarks to the Author):

The present paper describes a genome-wide association study of DNA-methylation with diabetes-relevant endpoints, which is of high scientific and clinical interest. The dimension of the study is comprehensive, with over 4,800 samples in the discovery study and 11,750 in the replication. The paper reports and replicates nine novel loci of interest. The statistical analysis is done according to established protocols and is convincing. Mendelian randomization was attempted, but results were largely negative, likely due to lack of power and suitable instruments. Overlap with gene expression data, also organ specific is provided, and the role of genetic variation is also addressed. In summary, the paper is solid, but remains rather descriptive and does not provide extensive new biological insights, above reporting the nine novel loci, which are, however, important results that merit publication in an important journal.

Minor points:

Comment on why the discovery was done in a much smaller cohort than the replication – some

hits may have been lost.

cg19693031 near TXNIP has been reported as a diabetes associated CpG site in multiple studies. Comment on why it did not appear here? Was the CpG site tested? Is there an association signal? Is it related to the fact that study participants were all non-diabetic?

Some associations depend on whether BMI is accounted for or not. A more profound analysis why this is the case would be interesting, maybe by relating results to the Wahl et al. EWAS with obesity (Nature 2016)?

Two out of the nine CpG sites apparently do not show consistent directions between ethnicities. More details on these two cases should be provided. Is the inconsistency significant?

Title: Novel DNA methylation sites of glucose and insulin homeostasis: an integrative cross-omics analysis (NCOMMS-18-27311)

Response to the editor:

“Your manuscript entitled “Novel DNA methylation sites of glucose and insulin homeostasis: an integrative cross-omics analysis” has now been seen by 2 referees. You will see from their comments below that while they find your work of interest, some important points are raised. We are interested in the possibility of publishing your study in Nature Communications, but would like to consider your response to these concerns in the form of a revised manuscript before we make a final decision on publication.”

We thank the editor for giving us the opportunity to address the comments from the referees. Overall, we included substantial changes in the manuscript, mainly on the interpretation of our findings and re-arranging the text so that the reader understands our reasoning. The main conclusions and findings have not been changed. We believe that these changes not only addressed the reviewer’s comments but also increased the quality of our manuscript. Please find below is the response of the reviewer’s questions in detail.

Response to the reviewers:

Reviewer #1 (Remarks to the Author):

“This is a large scale study which maps DNA methylation changes associated with fasting glucose and insulin in non-diabetic individuals. The study is based on the assumption that genetic variants drive methylation status of CpGs, which in turn regulate gene expression. Like in numerous other EWAS, only a small number of statistically significant CpG methylation differences were detected, and their functional roles are not clear.”

We agree with the reviewer that as up to date, only a limited number of methylation loci have been discovered and replicated for type 2 diabetes and related outcomes¹⁻⁷. In our current report, we doubled the number of CpG loci identified, which thus makes a substantial improvement upon our earlier knowledge. Of note is that we have been able to replicate our novel discoveries further. We regret that the functional impact of the EWAS and integration was not clear to the reviewer. We have restructured the manuscript and expanded the functional interpretation of the EWAS loci. This facilitates a better understanding of the functional relevance of our findings.

“1. The study is carefully performed and followed the well-known rules for epigenome-wide association studies, including mendelian randomization, and does not have any evident major limitations or shortcomings. The main problem is that it lacks innovation and new insight. Pages are full of gene names, p values, and other technical details which do not lead into any new understanding of the role of epigenetic factors in the phenotypes of interest.”

We are grateful that our methodology is considered to be solid and valid by the reviewer. As discussed above under the general comment, we have highlighted the new functional insight into the new version of the manuscript. We also underscore to the reader what are the new insights and in what way our work is novel. We have tried to reduce the technical details (gene names and p-values) but we cannot avoid all, particularly not in the discovery phase. We do explain the impact and insight beyond the p-values. This has indeed improved the readability of the manuscript.

“Only researchers working on glucose metabolism-related traits and diseases will benefit from this paper.”

We agree that the paper is of most interest to researchers working on glucose metabolism. This is an important field of research, clinically and scientifically. The prevalence of diabetes is rising due to the obesity

epidemic. Diabetes is a major health care problem, economically but also clinically with a significant risk of comorbidity. Disturbances in insulin-mediated glucose metabolism is a major driver of micro- and macrovascular complications including peripheral artery disease, but also kidney, eye and brain pathology (dementia) We are convinced that our research is also of interest to those who are not working on glucose metabolism as it yields a framework of integrating methylation research with genomics and transcriptomics.

“2. One evident and important question stemming from the study design, but not addressed/discussed by the authors, is the relationship between dynamic levels of glucose/insulin and cytosine methylation which assumed to be static in terminally differentiated somatic cells. The dozen or so cytosines detected under “fasting” conditions – would they also show an association with glucose/insulin after a meal? Or it will be a different set of cytosines whose methylation status correlate with high(er) levels of glucose/insulin?”

The reviewer raises a very interesting point. As is common practice in large epidemiological studies, we studied fasting glucose and insulin levels, to control for confounding by a recent diet that makes glucose levels incomparable between participants. In the current study, we cannot address the effect of these methylation marks on dynamic changes in glucose and insulin levels after a meal. We have included the point of the referee in the revised manuscript, in the discussion section, from line 343. "DNA methylation globally is considered a relatively stable epigenetic mark that can be inherited through multiple cell divisions^{8,9}. However, some changes can be dynamic reflected by recent environmental exposures. This phenomenon could be site-specific. While our study provides a snapshot of associations specific to the fasting state, instant methylation of different CpG sites in the vicinity of *IRS2* and *KDM2B* have been reported earlier¹⁰. Such effects may also occur at the loci presented in the present study."

“3. All findings are provided by Z scores and p values, which are not informative in terms of the actual DNA methylation differences and effect sizes. Since the final CpG set is quite small it would be feasible to show their profiles (i.e. scatterplot of correlation with both fasting insulin and glucose). Or to present the coefficients from the related models in an interpretable form (i.e. increase/decrease in fasting insulin per 1% of methylation). This would enhance the interpretability of results, enable comparisons between CpGs and their effects on insulin and glucose as well as provide insight into biological meaningfulness of the effect.”

We agree with the reviewer that the Z score does not reflect effect size. Therefore, we changed the Z scores in the tables to effect size (beta) which was calculated from the meta-analysis of the discovery cohorts. The beta can be interpreted as an increase/decrease in fasting glucose/insulin per 1% of methylation.

“4. It is not clear from the Methods if Houseman algorithm was used for correcting cell counts in the African population which was quite substantial (n=4355). The question is if the algorithm has been validated for non-Caucasian populations. Is it possible this was the reason that findings in European populations were not replicated in other ethnic groups (lines 165-168). “Consistent directions” is not a replication.”

The referee has a very careful remark. The BLSA study used measured cell proportions (granulocytes, lymphocyte, monocyte, and eosinophil) instead of the Houseman imputation when analyzing the African American ancestry samples. Indeed, as the referee rightfully suggests, the Houseman algorithm has been validated for non-Caucasians by Houseman and co-workers, albeit a small number of African-Americans are included in their original paper¹¹. The Houseman algorithm to impute cell counts has been widely used in the African population in previous EWAS papers¹²⁻¹⁵.

Although we cannot exclude that the African data are not comparable, we argue that the failure to replicate the association of the DNA methylation sites and fasting glucose/insulin in the non-Caucasian populations is more likely due to other reasons than the applied method to adjust for cell counts in the model. The major problem is the small size of the cohort resulting in a low statistical power to detect associations. In addition there

are genetic differences that hamper comparisons. This information is included in each cohort description in the supplementary notes.

“5. Bonferroni correction was used to adjust the p-values based on the number of CpGs but not applied for the number of models because the models are highly correlated. Are their correlations stable and do not change depending on age, BMI, some other parameters? E.g. how about insulin and glucose correlations in individuals with high BMI and insulin resistance?”

We understand the reviewer’s concern. Based on the comment, we checked the correlation of fasting insulin and glucose in different subgroups stratified by gender, BMI and insulin. Pearson’s correlation tests were used in the non-diabetic populations from the large population-based cohort, Rotterdam study. The results can be found in the table below. In all tests, the correlation coefficient is larger than 0.23 and is not different between males/females, normal/high BMI individuals or between individuals with normal or high insulin levels. Moreover, comparing our data with other multiple testing methods, e.g. FDR, Bonferroni test is a conservative method already^{16,17}. We are conservative, i.e., we aimed to minimize the probability of false positive findings.

Population	Sample size	Correlation coefficient	P-value
Overall	8226	0.35	1.6×10^{-235}
Female	4717	0.36	1.3×10^{-146}
Male	3509	0.33	2.0×10^{-90}
Normal BMI (< 25 kg/m ²)	2733	0.27	7.4×10^{-48}
Overweight (BMI ≥ 25 kg/m ² & BMI < 30 kg/m ²)	3898	0.27	1.0×10^{-67}
Obese (BMI ≥ 30 kg/m ²)	1595	0.29	1.8×10^{-32}
High insulin level (> 72 pmol/L)	3829	0.23	8.9×10^{-47}
Normal insulin level (≤ 72 pmol/L)	4397	0.23	9.2×10^{-55}
Age ≥ 70 years old	2264	0.35	5.5×10^{-65}
Age < 70 years old	5962	0.35	1.8×10^{-175}

“6. Editing is required. E.g. line 197 “were testes” – were tested; line 319 Linear mixed models were used in NTR and TwinsUK accounting_for_the family structure.”

Thanks for these comments. We double checked the whole manuscript again and corrected the typo and grammar mistakes.

Reviewer #2 (Remarks to the Author):

“The present paper describes a genome-wide association study of DNA-methylation with diabetes-relevant endpoints, which is of high scientific and clinical interest. The dimension of the study is comprehensive, with over 4,800 samples in the discovery study and 11,750 in the replication. The paper reports and replicates nine novel

loci of interest. The statistical analysis is done according to established protocols and is convincing. Mendelian randomization was attempted, but results were largely negative, likely due to lack of power and suitable instruments. Overlap with gene expression data, also organ specific is provided, and the role of genetic variation is also addressed. In summary, the paper is solid, but remains rather descriptive and does not provide extensive new biological insights, above reporting the nine novel loci, which are, however, important results that merit publication in an important journal.

We thank the reviewer for highlighting that our results are solid. We do agree that the Mendelian Randomization experiments may have lacked power. We point this out in the Discussion section. We do regret that it appears that we do not provide extensive new biological insights. We do realize we have failed to highlight these in the earlier version. We now have rewritten the manuscript with a focus on biological insight. Although our basis results and findings have remained unchanged, we added the functional and regulatory annotation of all the novel loci and we rewrote a substantial part of the manuscript interpreting the original results.

“Minor points: Comment on why the discovery was done in a much smaller cohort than the replication – some hits may have been lost.”

We would like to convince the reviewer on this: to replicate findings a larger cohort is needed than in the discovery. The reason for this is John Ioannidis winner’s curse: the hits in the discovery are often inflated ones (which are easy to find significant) and therefore a smaller effect is expected in the replication phase, which requires a larger samples size to discover. More importantly, the power of the discovery cohort is high. Using a sample size of 4,808 individuals, our discovery set has 80% power to detect an effect size (f^2) as small as 0.011 with a P-value 1.27×10^{-7} (current P-value threshold of EWAS)^{18,19}

“cg19693031 near TXNIP has been reported as a diabetes associated CpG site in multiple studies. Comment on why it did not appear here? Was the CpG site tested? Is there an association signal? Is it related to the fact that study participants were all non-diabetic?”

We thank the reviewer for pointing this out. Indeed, cg19693031 was tested in the current study but did not pass the p-value threshold (1.27×10^{-7}) which was applied for multiple testing in the current hypothesis-free design. However, the association of cg19693031 and fasting glucose in our study is still obvious (in BMI unadjusted model, p-value = 3.3×10^{-7} ; in BMI adjusted model, p-value = 7.6×10^{-7}). It barely missed the threshold – as pointed out to referee – we have been extremely conservative with the view to prevent false positive findings that cannot be replicated. Of note is that cg19693031 is not associated with fasting insulin (in BMI unadjusted model, p-value = 0.30; in the BMI adjusted model, p-value = 0.37). This is consistent with the previous study that cg19693031 is associated with glucose, HbA1c and diabetes^{4,5,7}. We added these findings in the discussion, from line 352: “Last but not least, cg19693031 in TXNIP has been repeatedly associated with type 2 diabetes case-control status earlier^{1,4,5}. Although it did not pass our predefined EWAS significance threshold, TXNIP is associated with fasting glucose in the non-diabetic population (P-value = 7.6×10^{-7} in the BMI adjustment model) if we take the current study aiming to replicate earlier findings.”

“Some associations depend on whether BMI is accounted for or not. A more profound analysis why this is the case would be interesting, maybe by relating results to the Wahl et al. EWAS with obesity (Nature 2016)?”

We agree with the reviewer’s comment that the role of BMI is worth clarification. We agree with the reviewer’s comment that the role of BMI is worth clarification. We related our results with Wahl et al.’s EWAS study and further studied the interplay between BMI, fasting glucose and insulin levels and differential methylation in the circulation. We complement that insulin metabolism is the key player underlying the fact reported by Wahl et al.’s study that the methylation patterns in blood predict future diabetes. Combining with Wahl et al.’s study, we have also shown that BMI may be a confounder of associations for some CpGs but may be in the causal pathway

for others. We have added this as a new part (Part 5) in the results and added a paragraph to discuss it (the third paragraph in the discussion section).

“Two out of the nine CpG sites apparently do not show consistent directions between ethnicities. More details on these two cases should be provided. Is the inconsistency significant?”

Thanks for the reviewer’s comment. The two CpG sites are cg18881723 in *SLAMF1* and cg13222915 in *1q25.3*. The directions in Hispanic ancestry is different from that in European and African ancestry, but the P-values are non-significant in Hispanic ancestry (P-value = 0.63 in cg18881723 and P-value = 0.092 in cg13222915). That implies we should see the effect in Hispanics as not different from 0. This is shown in Supplementary Table 3 and we also added the details in the result section, from line 171. “Two CpG sites (cg18881723 and cg13222915) show the opposite direction for the effect estimate in HA ancestry population as compared to the other two populations. However, the estimates of effect size are not significantly different from zero (P-value = 0.63 in cg18881723 and P-value = 0.092 in cg13222915).”

References

- 1 Chambers, J. C. *et al.* Epigenome-wide association of DNA methylation markers in peripheral blood from Indian Asians and Europeans with incident type 2 diabetes: a nested case-control study. *Lancet Diabetes Endocrinol* **3**, 526-534 (2015).
- 2 Al Muftah, W. A. *et al.* Epigenetic associations of type 2 diabetes and BMI in an Arab population. *Clin Epigenetics* **8**, 13 (2016).
- 3 Kulkarni, H. *et al.* Novel epigenetic determinants of type 2 diabetes in Mexican-American families. *Hum Mol Genet* **24**, 5330-5344 (2015).
- 4 Florath, I. *et al.* Type 2 diabetes and leucocyte DNA methylation: an epigenome-wide association study in over 1,500 older adults. *Diabetologia* **59**, 130-138, (2016).
- 5 Soriano-Tarraga, C. *et al.* Epigenome-wide association study identifies TXNIP gene associated with type 2 diabetes mellitus and sustained hyperglycemia. *Hum Mol Genet* **25**, 609-619, (2016).
- 6 Yuan, W. *et al.* An integrated epigenomic analysis for type 2 diabetes susceptibility loci in monozygotic twins. *Nat Commun* **5**, 5719, (2014).
- 7 Walaszczyk, E. *et al.* DNA methylation markers associated with type 2 diabetes, fasting glucose and HbA1c levels: a systematic review and replication in a case-control sample of the Lifelines study. *Diabetologia* **61**, 354-368, (2018).
- 8 Bird, A. DNA methylation patterns and epigenetic memory. *Genes Dev* **16**, 6-21, (2002).
- 9 Kim, M. & Costello, J. DNA methylation: an epigenetic mark of cellular memory. *Exp Mol Med* **49**, e322, (2017).
- 10 Rask-Andersen, M. *et al.* Postprandial alterations in whole-blood DNA methylation are mediated by changes in white blood cell composition. *Am J Clin Nutr* **104**, 518-525, (2016).
- 11 Houseman, E. A. *et al.* DNA methylation arrays as surrogate measures of cell mixture distribution. *BMC Bioinformatics* **13**, 86, (2012).
- 12 Wang, X. *et al.* An epigenome-wide study of obesity in African American youth and young adults: novel findings, replication in neutrophils, and relationship with gene expression. *Clin Epigenetics* **10**, 3, (2018).
- 13 Demerath, E. W. *et al.* Epigenome-wide association study (EWAS) of BMI, BMI change and waist circumference in African American adults identifies multiple replicated loci. *Hum Mol Genet* **24**, 4464-4479 (2015).
- 14 Wahl, S. *et al.* Epigenome-wide association study of body mass index, and the adverse outcomes of adiposity. *Nature* **541**, 81-86 (2017).
- 15 Marioni, R. E. *et al.* Meta-analysis of epigenome-wide association studies of cognitive abilities. *Mol Psychiatry* **23**, 2133-2144, (2018).

- 16 Narum, S. R. Beyond Bonferroni: less conservative analyses for conservation genetics. *Conservation genetics* **7**, 783-787 (2006).
- 17 García, L. V. Escaping the Bonferroni iron claw in ecological studies. *Oikos* **105**, 657-663 (2004).
- 18 Kutner, M., Nachtsheim, C. J., Neter, J. & Li, W. Applied linear statistical models. McGraw-Hill. *New York* (2005).
- 19 Ryan, T. P. *Sample size determination and power*. (John Wiley & Sons, 2013).

REVIEWERS' COMMENTS:

Reviewer #2 (Remarks to the Author):

The authors have responded to most of my concerns. I just have a few minor points:

"We would like to convince the reviewer on this: to replicate findings a larger cohort is needed than in the

discovery." ... I agree that due to the winner's curse effect sizes may be weaker in the replication study. However, also the targeted level of significance is orders of magnitude smaller in a replication. Anyway, I am not asking the authors to rerun everything on the larger dataset to get a few more hits. I expect that in the future these cohorts will likely participate in some kind of a meta-analysis, so there will be plenty opportunity to uncover more hits later.

TXNIP: I suggest adding also the following line (from the rebuttal) to the main text: "Of note is that cg19693031 is not associated with fasting insulin (in BMI unadjusted model, p-value = 0.30; in the BMI adjusted model, p-value = 0.37)."

Data availability: Google Drive is not a persistent depository for research data. I suggest providing the summary statistics either as Supplemental Tables (if Nat Comm policy permits), or alternative to a suitable database (Nature Data has a list).

Response to referee

Reviewer #2 (Remarks to the Author):

The authors have responded to most of my concerns. I just have a few minor points:

“We would like to convince the reviewer on this: to replicate findings a larger cohort is needed than in the discovery.” ... I agree that due to the winner’s curse effect sizes may be weaker in the replication study. However, also the targeted level of significance is orders of magnitude smaller in a replication. Anyway, I am not asking the authors to rerun everything on the larger dataset to get a few more hits. I expect that in the future these cohorts will likely participate in some kind of a meta-analysis, so there will be plenty opportunity to uncover more hits later.

TXNIP: I suggest adding also the following line (from the rebuttal) to the main text: “Of note is that cg19693031 is not associated with fasting insulin (in BMI unadjusted model, p-value = 0.30; in the BMI adjusted model, p-value = 0.37).”

Data availability: Google Drive is not a persistent depository for research data. I suggest providing the summary statistics either as Supplemental Tables (if Nat Comm policy permits), or alternative to a suitable database (Nature Data has a list).

We thanks for the referee’s support on our paper. We have added the sentence in the main text and uploaded the data in a proper database.